# Cold spells over Greenland during the mid-Pliocene Warm Period

I. M. C. Sousa ⬤ [1] ✉, C. Hillaire-Marcel ⬤ [1], A. de Vernal ⬤ [1],
J. -C. Montero-Serrano[2] & A. M. R. Aubry[1]

The mid-Pliocene Warm Period (mPWP; 3.26–3.02 Ma) is an interval often suggested as a potential analogue of the near future climate and fate of the Greenland Ice Sheet (GIS). Here, neodymium and lead isotopes from marine sediment cores collected off the southern Greenland margin suggest pulses of intense glacial erosion of Precambrian terranes during this interval, while grain size data indicate a reduction in the strength of contour currents, both following a near obliquity cycle (~41 ka) pacing. These cold spells were thus sufficiently intense to trigger recurrent ice growth over Greenland, even under the high atmospheric $CO_2$ concentration (~400 ppmV) of the interval, before the intensification of the Northern Hemisphere Glaciation (~2.7 Ma). However, the mPWP was marked by a low amplitude in the axial tilt oscillation of the Earth, thus lesser variations in summer insolation at high latitudes than in the present era. Therefore, although it may offer some similarities with the future of the Earth's climate, the mid-Pliocene cannot be seen as a genuine analogue for predicting the fate of the GIS.

The current atmospheric $pCO_2$ is within estimates for the mid-Pliocene Warm Period (mPWP; 3.264–3.025 Ma)[1–4]. With atmospheric $pCO_2$ ranging from $360 \pm 80$ ppmv[5] to $530 \pm 110$ ppmv[6] and the boreal forest reaching the northernmost Arctic Canada[7,8], the mPWP is often seen as a possible analogue of the near future of our planet[2], worth investigating as it depicted a land-ocean distribution and an ocean bathymetry similar to those of the present-day[1,9]. Modelling studies indicate that during the mPWP, the Greenland Ice Sheet (GIS) extent may have been considerably reduced[10], thus possibly contributing to ~15 m higher than present sea-level[11]. Data and model simulations suggest sea surface temperatures (SST) of at least +3 °C in Arctic regions during this interval[1], and terrestrial Arctic temperatures 11 to 16 °C warmer than present[8]. Diverse scenarios have been put forth regarding the extent of the GIS during the mPWP. These scenarios range from a substantial 50% reduction in the size of the GIS[12] to a situation where ice caps were confined to the elevated areas of south/southeastern and eastern/northeastern Greenland[13,14] (Fig. 1A, B). Some models suggest that the maximum ice extent occurred in northeast Greenland[14,15]

(Fig. 1C). Still, proxies documenting the mPWP Greenland ice cover are few, and mostly indirect as they are restricted to information from marine records off Greenland margins[16].

In this study, we use a sedimentary record from the Integrated Ocean Drilling Program (IODP) on the Eirik Drift off southern Greenland (site U1307; Fig. 1) to indirectly assess the dynamics of the GIS during the mPWP. The Eirik Drift is a contourite drift moulded by the Western Boundary Undercurrent (WBUC)[17], and its formation can be traced back to the Miocene[18]. Previous paleoceanographic studies suggest that the strength of the WBUC was significantly reduced during glacial intervals but exhibited high flow rates notably during Quaternary interglacials[19–23].

Ice erosion of Precambrian terranes in Southern Greenland has been shown to deliver sediments of singular radiogenic isotope compositions to the Eirik Drift during the Pleistocene and Holocene[19,24]. Here, we use the isotope signature of the Precambrian source rocks to document changes in glacial activity over southern Greenland during the mPWP, as recorded in sediment from IODP site U1307. This site was

[1]Geotop and Département des sciences de la Terre et de l'atmosphère, Université du Québec à Montréal (UQAM), Montréal, QC, Canada. [2]Institut des sciences de la mer (ISMER), Université du Québec à Rimouski (UQAR), Geotop & Québec-Océan, Rimouski, QC, Canada. ✉ e-mail: moreno_cordeiro_de_sousa.isabela@uqam.ca

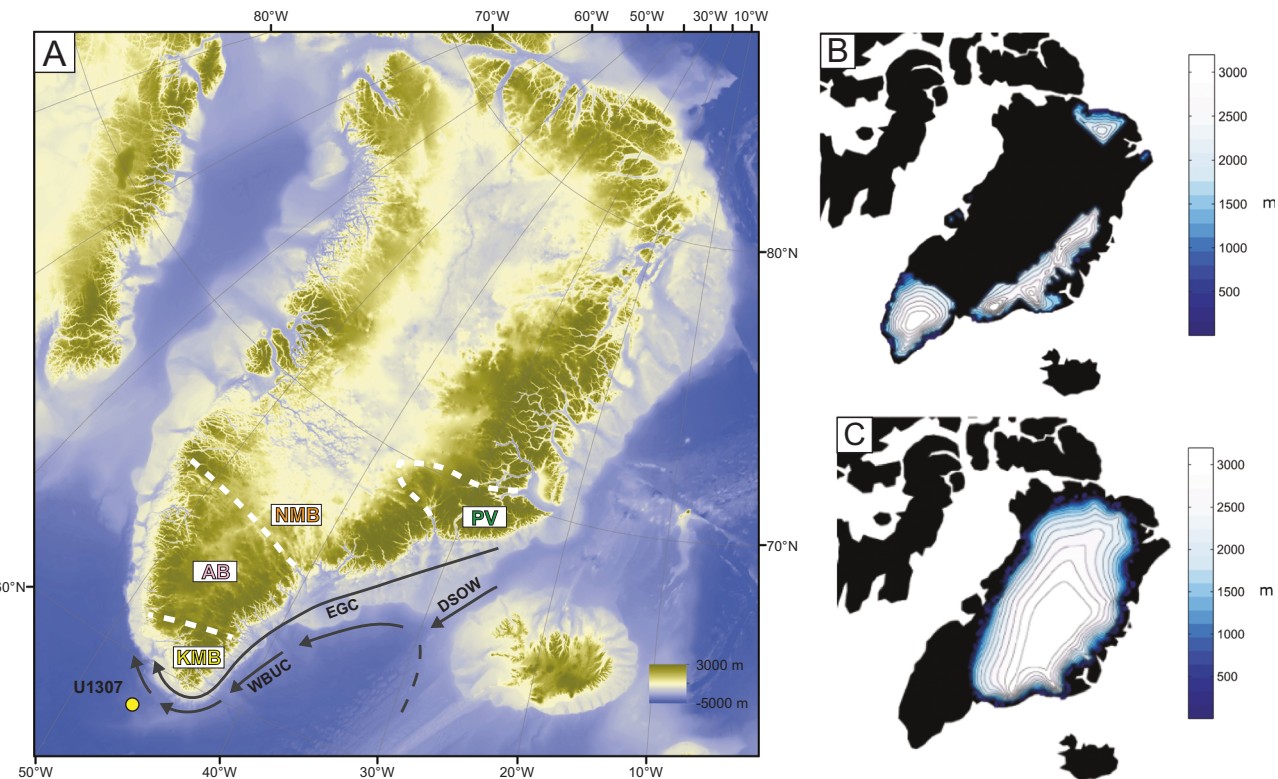

**Fig. 1 | Greenland topography[80], bedrock terranes, and ice sheet surface elevation during the mid-Pliocene Warm Period. A** Greenland topography[80] and boundaries of major bedrock terranes in Greenland and Iceland[29]. AB: Archean Block $\varepsilon_{Nd(0)}$ ~ −32.4, $^{207}Pb/^{206}Pb$ ~ 0.94, KMB: Ketilidian Mobile Belt ($\varepsilon_{Nd(0)}$ ~ −20.7, $^{207}Pb/^{206}Pb$ ~ 0.74), NMB: Nagssugtoqidian Mobile Belt ($\varepsilon_{Nd(0)}$ ~ −30.5, $^{207}Pb/^{206}Pb$ ~ 0.90), and PV: Paleogene volcanism ($\varepsilon_{Nd(0)}$ ~ 4.5, $^{207}Pb/^{206}Pb$ ~ 0.86)[19]. WBUC: Western boundary undercurrent. DSOW: Denmark Strait Overflow Water. EGC: East Greenland Current. **B** Ice sheet surface elevation under minimal ice cover during the Pliocene from GRISLI model[14,15]. **C.** Ice sheet surface elevation under minimal ice cover during the Pliocene from GLIMMER model[14,81].

investigated by several authors who set up the first age models[20–22]. At this site and along the trajectory of the WBUC, grain-size data analyses of surface sediments show that silt overweighs the clay size fraction that may originate from remote sources, north of the Denmark Strait (Fig. 1). Notwithstanding the dominance of silt, the effective grain size of the sediment is sensitive to changes in the strength of the WBUC[23–25]. Disregarding the content of ice rafted debris (IRD)[22], this behaviour results in the deposition of coarser sediments during interglacials due to high shear stress and winnowing[26]. Whereas sediments originating from the Canadian margin are transported southward through the Labrador Current[27], the Eirik Ridge sediments mostly relate to proximal terrestrial source supplies from southeastern Greenland, as documented in Quaternary records[19,24].

The terrestrial sediment sources from southeastern Greenland terranes include the Archean Block (AB), the Paleoproterozoic Ketilidian Mobile Belt (KMB), the Paleoproterozoic Nagssugtoqidian Mobile Belt (NMB), and the Paleogene volcanism (PV) that also formed Iceland[28,29] (Fig. 1). Each of these terranes holds distinctive neodymium (Nd) and lead (Pb) isotope compositions[19,24], as detailed in the supplementary material, thus supplying sediments with distinctive chemical signatures of their source to IODP site U1307 (Fig. 2).

For the documenting of the mPWP interval, we used cores from the A and B holes of IODP site U1307 (58°30′N, 46°24′W, 2575 m water depth; Fig. 1)[30] to set a continuous Nd and Pb isotope record throughout the mPWP interval. Earlier studies of U1307, notably by Channel et al.[30], Sarnthein et al.[20], Mazaud et al.[21], Aubry et al.[31] and Blake-Mizen et al.[22] provided detailed features of paleoceanographic changes at this site, with a chronostratigraphy based on oxygen isotope stratigraphy and paleomagnetic reversal age constraints. Here, we used the most recent age model of Blake-Mizen et al.[22] for

investigating the 3.22–2.95 Ma interval, i.e., broadly the mPWP episode, with a temporal sampling resolution of ~5 ka. In addition to Nd and Pb isotope measurements (here expressed as $\varepsilon_{Nd(0)}$ and $^{207}Pb/^{206}Pb$; methods in supplementary material) performed on the <63 μm sediment fraction, we used calcite contents derived from quantitative X-ray diffraction analysis of bulk sediments and laser measurements of mean grain size, sand contents and sortable silt mean size ($SS_{mean}$). X-ray-based calcite abundances are used as tracers of biogenic production[32], while the coarseness in $SS_{mean}$ and increased sand content are used as indicators of bottom current strength[33,34]. Details about analytical methods are to be found in the supplementary material. Our main results are illustrated in Fig. 3, together with the $\delta^{18}O$ stack of Lisiecki and Raymo[35], and the U1307 $\delta^{18}O$ measurements of the planktic foraminifera *Neogloboquadrina atlantica*, by Sarnthein et al.[20] for potential comparison with other site records.

## Results

The mPWP sedimentary record at Eirik Drift exhibits recurring $\varepsilon_{Nd(0)}$ excursions of −2 to −7 units from background values, with the most radiogenic values ranging from −15 to −11. These excursions are in phase with $^{207}Pb/^{206}Pb$ upsurges from a background value of ~0.84 toward values exceeding 0.88. As documented for Quaternary glacial intervals in several studies[19,24], such unradiogenic Nd/radiogenic Pb excursions record enhanced terrigenous supplies from the old Precambrian craton of southern Greenland (Fig. 3). We will thus now refer to these excursions as "glacials", and to the interval between them as "interglacial" stages within the mPWP time window.

The overall $\varepsilon_{Nd(0)}$ and $^{207}Pb/^{206}Pb$ signatures depict trends across the mPWP interval (Fig. 3). The $\varepsilon_{Nd(0)}$ data from both the glacial and interglacial sediments follow a trend towards more radiogenic values

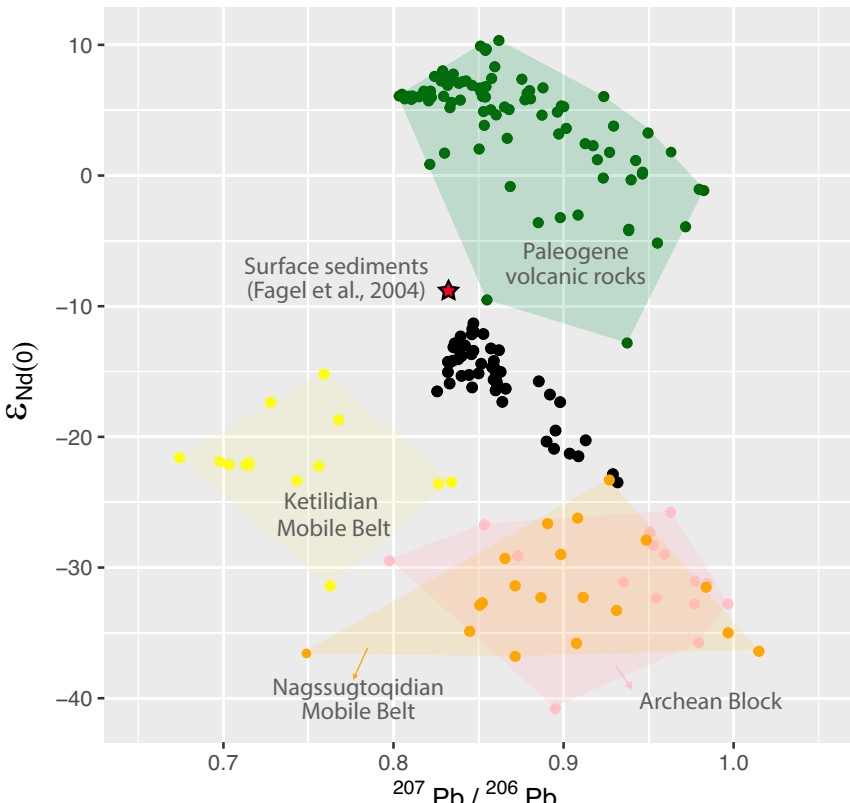

**Fig. 2 | $^{207}$Pb/$^{206}$Pb vs $\varepsilon_{Nd(0)}$ composition of bedrock terranes in Greenland and Iceland.** Black dots are samples from site U1307 analyzed in this paper AB: Archean Block (pink), KMB: Ketilidian Mobile Belt (yellow), NMB: Nagssugtoqidian Mobile Belt (orange), and PV: Paleogene volcanism (green)[19]. Surface sediment composition (red star) from Fagel et al. (2004)[36].

through time. The $\varepsilon_{Nd(0)}$ values of interglacial sediments (−15 to −11) suggest an almost even mixing of three end-members (Fig. 2): PV, KMB, and NMB/AB. These values remain significantly less radiogenic than those recorded by Fagel et al.[36] in late Holocene and surface sediment clays from the area ($\varepsilon_{Nd(0)=}$−9.3)[36] (Fig. 3). The more radiogenic recent values indicate higher supplies from the Paleogene volcanic sediments[36], outcropping on both sides of Denmark Strait (Fig. 1). The $^{207}$Pb/$^{206}$Pb ratios of interglacials are within the range of late Holocene sediments[36]. The glacial $^{207}$Pb/$^{206}$Pb excursions depict a trend indicating increasing contributions from less radiogenic sources towards the younger part of the record[37,38] (Fig. 3).

The Nd-Pb isotopic excursions are preceded by peaking calcite abundance, of variable amplitude, as well as by high mean grain size, sortable-silt size (SS$_{mean}$) and sand contents (Fig. 3). These features broadly highlight two distinct sedimentary regimes linked to the strength of the WBUC. One is related to strong currents during warm intervals, in particular during their middle to late part[32]. The other is related to low flow rates during cold intervals. The interglacial layers illustrated in Fig. 3 record an early stage with relatively fine material, followed by a late stage marked by sediment coarsening, which is a feature observed during the present interglacial in the core site vicinity[32]. The quasi-absence of dolomite leads to the inference that calcite peaking values record enhanced biogenic carbonate fluxes during the late part of the interglacial intervals. A similar feature characterises the present interglacial[32].

Estimating changes in sediment accumulation rates between the two contrasted sedimentary regimes would require estimating the duration of the glacials vs interglacials. However, linear interpolations between age anchors cannot provide robust duration estimates due to highly variable sediment accumulation rates between and during both intervals[18,39]. Nevertheless, the recurrence of six, possibly seven (Fig. 3)

isotopic excursions over the 2280−3220 ka interval studied (grey stripes in Fig. 3) would be compatible with a mean duration of ~40 ka for each cold-warm cycle. The ~40 ka-periodicity is not clearly expressed when using the Blake-Mizen et al.[22] age model (supplementary material, Fig. S7). This can be due to both uncertainty of anchor ages, in particular those of paleomagnetic reversals, and biases linked to linear interpolation between these ages. Assigning the glacial pulses to specific glacial stages defined by the LR04 stack[35] throughout the interval constrained by the paleomagnetic anchor ages (supplementary material, Fig. S5) could be a way to define an age model, which would fit better with a ~40 ka cycle forcing (supplementary material, Fig. S5).

## Discussion

In the studied interval, cold spells correspond to ice growth pulses, with a possible ~40 ka pacing still to be demonstrated. These cold intervals also correspond to a weak WBUC, as suggested by finer grain size and low SS$_{mean}$ values (Fig. 3). Adding to their very low biogenic carbonate content, these glacials likely experienced low sediment accumulation rates over Eirik Drift. Such features result in problematic linkages between sediment thickness and time when using linear interpolation between anchor dates for the setting of an age model. Nonetheless, the decreasing trend defined by the SS$_{mean}$ record throughout the mPWP (Fig. 3) indicates an evolution toward a weaker WBUC, thus toward the near collapse of the Atlantic Meridional Overturning Circulation (AMOC)[40], during the more extreme glacial intervals of the Quaternary[41].

Conversely, the predominance of coarser sediments and high SS$_{mean}$ values recorded during warm stages, and notably during the latest warm stages of the mPWP sequence, unveil a higher flow rate of the WBUC (Fig. 3). Given the practically identical grain size parameters

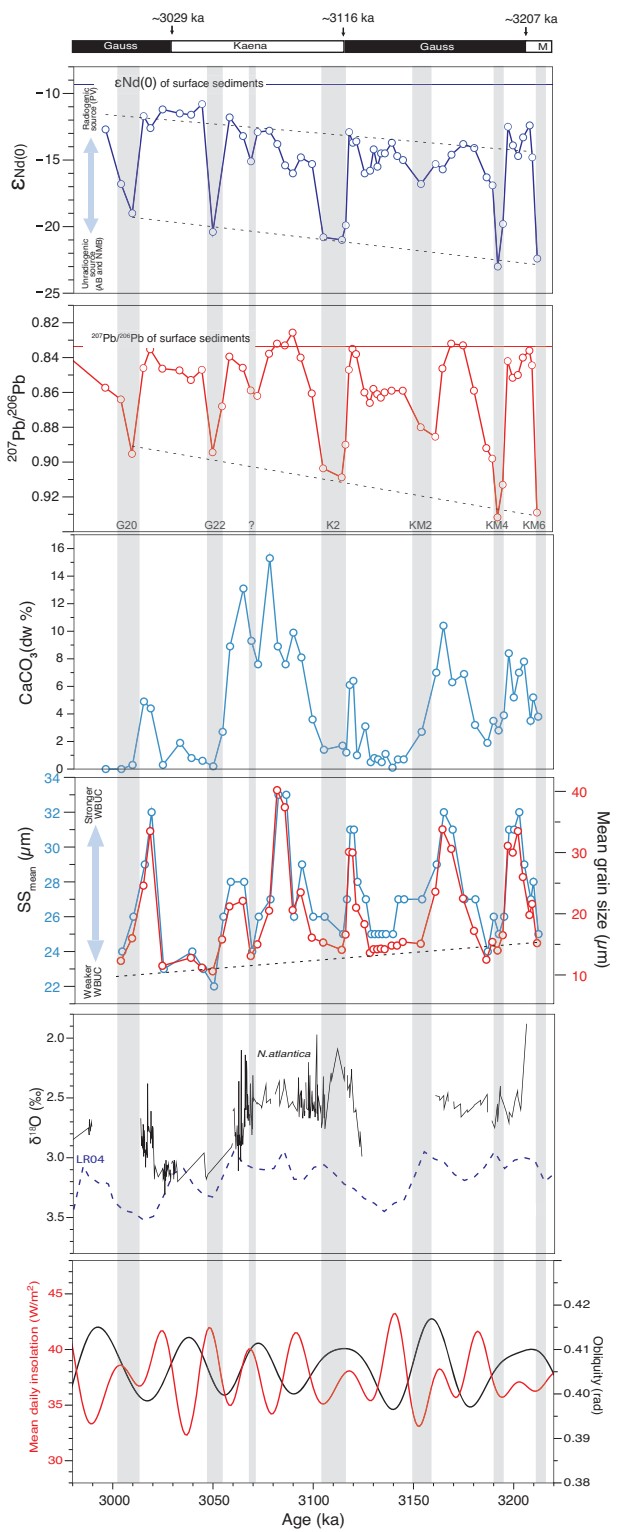

**Fig. 3 | Analyses on the sedimentary record of the mPWP (mid-Pliocene Warm Period).** From top to bottom; Paleomagnetic record of core U1307[39] (M = Mammoth) and approximate age of reversals[39,82]; Sediment $\varepsilon_{Nd(0)}$ values; Sediment $^{207}Pb/^{206}Pb$ ratios; Calcite abundance (%); $SS_{mean}$ (mean size of sortable silts, μm) and bulk mean grain size (μm); $\delta^{18}O$ of *N.atlantica*[20] samples vs the benthic LR04[35] stack; Earth's obliquity and mean daily isolation at -65°N, 314°E[49]. Grey bars highlight the glacial intervals over Southern Greenland. Long-term trends throughout the mPWP interval ($\varepsilon_{Nd(0)}$ glacials and interglacials, $^{207}Pb/^{206}Pb$ glacials, sortable silts and grain size) are highlighted by black dashed lines. The assignment of glacial layers to specific glacial stages of the sequence has been made using the lead isotope mid-transitions between warm and cold intervals, with reference to the chronology of Lisiecki and Raymo (2005)[35] carried into the sequence using the paleomagnetic reversal anchors (Supplementary Material Fig. S5).

(Fig. 1). As mentioned above, the sediments deposited during interglacials bear a stronger contribution from the PV source area and are thus characterised by a highly radiogenic Nd and unradiogenic Pb supplies. There is a long-term trend, from ~3220 to ~2980 ka (Fig. 3), in the Nd isotope compositions of both glacial and interglacial units, toward more radiogenic Nd endmembers, respectively from −22 to −19 (glacials) and from −14 to −11 (interglacials) $\varepsilon_{Nd(0)}$ values. In contradistinction, only the glacial $^{207}Pb/^{206}Pb$ ratios record a trend throughout the mPWP interval, in this case, toward less radiogenic values (from ~0.93 to ~0.88). This suggests a shift in the relative supplies of sediment sources, from the older to the younger terranes. Combining the isotope trends with the grain-size trend suggesting a weakening WBUC flow from one glacial to the next[23,45], one may infer increasing PV terrane erosion through the mPWP interval glacials, labelled by increasing radiogenic Nd signatures due to ice expansion in the eastern mountains (Fig. 1). Thus, throughout the mPWP, the southeastern Greenland ice mass occupying KMB/AB terranes expanded progressively northward from one glacial pulse to the next, possibly merging with ice from the northeastern mountainous PV/NMB area (Fig. 1B). Consequently, there is a clear trend of increasing overall contribution of the PV terrane towards the upper part of the record. During the cold spells, enhanced ice erosion mainly released detrital supplies from the Archean Block (AB) and Nagssugtoqidian Mobile Belt (NMB) (Figs. 1B and 2). Conversely, during interglacials, ice erosion was likely restricted to the mountainous KMB-AB area of southeastern Greenland (Figs. 1B and 2). This would explain the narrow cluster of Pb and Nd isotope compositions of interglacial detrital supplies between three endmembers (KMB, AB/NMB and PV): the KMB and AB/NMB endmembers, linked to ice erosion, and the PV endmember linked to increased WBUC supplies from the Denmark Strait (Fig. 2).

Assessing the effective orbital forcing of the glacial pulses over Greenland during the mPWP requires an age model with a suitable time resolution to document potential linkages with the obliquity cycle. Such a resolution cannot be achieved using paleomagnetic reversal ages, and the isotope stratigraphy record of U1307 during the interval has proven equally problematic[20,39]. The oxygen isotope data of planktic foraminifera initially reported by Sarnthein et al.[20] show shifts from ~2.5‰ (vs VPDB), i.e., close to the mid-to-late Holocene isotopic composition of *N. pachyderma*[32], in the same area[46], to heavier values (~3.1‰) (Fig. 3). This +0.6‰ shift is much smaller in amplitude than the ~2‰ glacial/interglacial offsets recorded during the Quaternary[32], thus pointing to a lesser amplitude in the ice volume variations during the mPWP. From a chronostratigraphic perspective, several issues can be raised about the $\delta^{18}O$ record of the mPWP at IODP Site U1307, as discussed by Sarnthein et al.[20], who proposed two possible age models, then by Blake-Mizen et al.[22], who proposed another one. None of the models proposed is totally satisfactory. Firstly, in the upper part of the record, these age models imply large amplitude changes in sedimentation rates without direct relation with the glacial/interglacial oscillations as documented from the grain-size parameters, which we associate with the effective variability of the WBUC. Secondly, there is

from one interglacial to the next, one may thus infer the return to nearly similar WBUC strength each time. As the duration of glacials remains unascertained, the duration of interglacial intervals is difficult to estimate, in part due to variable surface conditions[42,43] generating uneven fluxes of biogenic linked to coccolithophorid productivity[44] (Fig. 3).

The sources of eroded terrestrial materials deposited during the mPWP glacials differ from those of interglacials. Glacial units are characterised by highly radiogenic Pb and unradiogenic Nd-bearing supplies (Fig. 3) mainly derived from the AB and NMB source areas

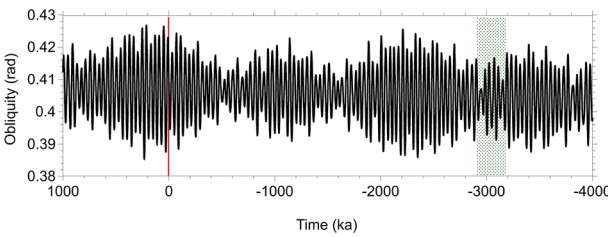

**Fig. 4 | Earth's obliquity during the last 4 Ma and the future 1 Ma[49].** The mPWP (mid-Pliocene Warm Period) occurred during an interval marked by low amplitude obliquity changes highlighted in green.

no unequivocal relationship between the heavier $\delta^{18}O$ values of *N. atlantica* at Site U1307 and glacial intervals inferred from all other parameters illustrated (Fig. 3).

Nonetheless, the number and relative distribution of the glacial peaks over the studied interval suggest some linkage with the ~41 ka obliquity cycle, as observed in Antarctica for the same interval[47,48]. However, at site U1307, this frequency does not stand out clearly in a periodogram when using the Blake-Mizen et al.[22] age model (supplementary material) as misfits between glacial pulses and low obliquity values[49] are noticeable. Assuming that the age model of Blake-Mizen et al.[22] is broadly correct, these misfits could relate to (i) a chronological resolution insufficient to resolve the obliquity cycle forcing due to the high variability in sedimentation rates, (ii) diachronous global benthic (LR04) vs. local planktic foraminifer $\delta^{18}O$ responses, especially during deglaciations[50–52]. Nevertheless, as documented in the supplementary material, we tentatively correlated the glacial intervals of U1307 defined from lead-isotope excursions towards unradiogenic values, to the glacial stages of the mPWP interval of LR04, as defined by Lisiecki and Raymo (2005)[50] (Fig. S4). Small offsets with the Blake-Mizen et al.[22] are observed but do not exceed a few thousand years. Fig. S4 also clearly illustrates the lesser mean sedimentation rates of glacial vs interglacial stages.

Looking deeper into the potential linkage between the cold spells of the mPWP and the obliquity cycle, one should note that the interval was characterised by obliquity oscillations of reduced amplitude (Fig. 4), thus by generally lower overall insolation at high latitudes[53]. That would allow for some ice growth, as suggested by O'Neill and Broccoli[48]. Such low-obliquity intervals are marked by an approximately 1.2 Ma maximum frequency and have been associated with "third order eustatic cycles and Neogene glaciations" by Lourens and Hilgen (1997)[54]. A decrease in Earth's axial tilt results in reduced annual insolation at high latitudes but in an increase at the equator[53]. Such changes in insolation gradients can generate atmospheric circulation shifts influencing moisture transport and continental snowfall. In the Northern Hemisphere, for example, a reduction in obliquity enhances summer half-year snowfall by 78% and winter half-year snowfall by 22%[53]. Now, as such low-amplitude obliquity oscillations will not be encountered within the next million years (Fig. 4), one cannot consider the mPWP as a close analogue of the near future of the Earth climate and, more specifically, of future ice conditions over Greenland.

Overall, a closer look at the potential relationship between the geochemical and sedimentological parameters with the obliquity cycles of the mPWP interval, shows significant offsets, particularly before 3080 ka (Fig. 3). We have seen that neither the $\delta^{18}O$ record nor the paleomagnetic record can provide time anchorages with a millennial time resolution. We are thus left with an overall time frame for the mPWP glacial pulses, but their assignment to specific insolation and obliquity cycles within the interval is currently based on hypothetical correlation using the available data.

Despite the high $pCO_2$ concentrations of the mid-Pliocene Warm Period (~400 ppmV), insolation changes related to orbital parameters (obliquity) were thus sufficient to trigger ice growth over Greenland

prior to the extensive Northern Hemisphere glaciations of the Quaternary, as already documented in earlier studies[16,41,55,56].

Summing up and adding to previous investigations at IODP site U1307[20,22,30,31] our isotopic and sedimentological analyses point to ice-growth pulses with pacing compatible with the ~41 ka obliquity cycle. As also observed during the Quaternary[23,57], the glacial/interglacial cycles of the mPWP were marked by significant changes in the strength of the Western Boundary Undercurrent. WBUC shows a significant flow reduction during the glacial pulses, and a long-term trend toward the Quaternary era, marked by a progressively decreasing WBUC from one glacial pulse to the following. This longer-term trend could have been driven by the $pCO_2$ changes of the Plio-Pleistocene transition[13].

Nd and Pb isotope compositions provided critical information about the source areas of the glacially eroded sediments. They mainly originated from the Precambrian basement of southern Greenland and from Paleogene volcanic terranes. A longer-term trend toward more radiogenic Nd and less radiogenic Pb source rocks points to an ice extension over eastern Greenland mountains from one glacial stage to the next, yielding progressively enhanced erosional supplies from the Paleogene volcanic terrane towards the Quaternary. These results imply enhanced spreading through time of the ice cover of the southern and eastern Greenland reliefs[58], merging over south Greenland, and even with some northeast ice cover. Such a scenario of the GIS evolution during the mPWP is more in agreement with results from numerical experiments of minimal ice cover from the GRISLI model (Fig. 1B), than with those of the GLIMMER MODEL (Fig. 1C). Our findings are not compatible either with a constant small ice cap configuration throughout the Pliocene proposed by Solgaard et al.[58].

During the mPWP, both over Greenland, as shown here, and Antarctica[47,59], Earth's obliquity has apparently been a major driver of ice growth pulses. While several works refer to the mPWP as a "close" analogue for the future climate conditions of the Earth[2,9], the reduced amplitude in the axial tilt oscillation of the Earth, and consequent low overall summer insolation conditions at high latitudes will not be encountered within the next few hundred thousand years. Therefore, if offering analogies with the near future of the Earth's climate, the mPWP interval cannot be seen as a genuine analogue specifically for predicting the fate of the GIS. Adding that the present $pCO_2$ has already exceeded that of the late Pliocene, perhaps, from a global perspective, the Eocene should now receive more attention[9], although it cannot be seen either as a reliable analogue with respect to Greenland ice cover, due to the significant paleogeographic differences between the Eocene and the present[60].

## Methods
### Grain size measurements
Sediment grain size analyses (<2 mm fraction) were performed using a Malvern-Panalytical Mastersizer 3000™ particle-size analyzer equipped with a Hydro LV module following the instrumental conditions outlined in Belzile and Montero-Serrano (2022)[61], at the Institut des sciences de la mer (ISMER) of University of Québec in Rimouski (UQAR). Before analysis, sediment samples were pretreated with 10 mL of hydrogen peroxide ($H_2O_2$; 30% v/v) to remove organic matter. Carbonates were not removed from sediment samples, as they provide information on primary production and sediment provenance[62]. Sediment samples were diluted with ~20 mL of sodium hexametaphosphate, sieved at <2 mm, and disaggregated using an in-house rotator for 12 h prior to particle size measurements. The grain size data obtained were processed using the GRADISTAT™ software version 9.1[63]. For the documenting of deep current velocities, the sortable silt means size ($SS_{mean}$), which corresponds to the mean grain size of the non-cohesive silt fraction (10−63 μm) and the percentage of sortable silt (SS%) in the <63 μm fraction[33] were calculated using the approach developed by McCave and Andrews (2019)[34]. Results are presented in Supplementary Datasets 1 and 2.

## Determination of calcite and dolomite by quantitative X-ray diffraction

Calcite and dolomite were determined by quantitative X-ray diffraction (qXRD) at ISMER-UQAR following the method developed by Eberl (2003)[64]. For these analyses, ~1 g of each sample (<2 mm fraction) was spiked with 0.25 g of corundum and then ground in a McCrone™ micronizing mill using 5 mL of ethanol to obtain a homogenous powder. The slurry was oven dried in a hood and then slightly homogenized with an agate mortar. Next, 0.5 mL of Vertrel™ solvent was added to the mixture to prevent the possible agglomeration of finer particles. The powder sample was then sieved (<500 μm), side-loaded into the holders and analyzed on a PANalytical X'Pert™ powder diffractometer. Samples were scanned from 5° to 65° 2θ in steps of 0.02° 2θ, with a counting time of 2 s per step. For the quantification of the calcite and dolomite, sediment XRD scans obtained were converted into mineral weight percent (wt. %) using the R package "powdR"[65]. powdR uses a full pattern summation approach that permits the quantification of whole-sediment mineralogy with an average absolute bias of 0.6% for non-clay minerals[65]. The quality of the fitting procedure was evaluated for the R-weighted profile (rwp; best least-squares fit between observed and calculated model). The rwp for the samples analyzed ranged from 0.17 to 0.22 (median 0.18). An rwp value between 0.2 and 0.3 indicates a good fit for geological samples[66]. Results are presented in Supplementary Dataset 3.

## Nd and Pb isotope measurements

Chemical separation and Nd-Pb isotopic analyses were conducted at Geotop-UQAM. Prior to analysis, 1 g of each sample was wet sieved through a 64-μm Nitex™ mesh and then dried at the hotplate (~80 °C). Silt was not separated from the clay because grain size analyses showed that clay constitutes only 10% of the samples. In a very conservative estimate, in which clays from the cold spells would have been entirely derived from Precambrian sources ($^{143}Nd/^{144}Nd = 0.510978$)[19], their presence would account for a maximum increment of ~ -1 $\varepsilon_{Nd(0)}$ units, meaning that excursions would still be distinguishable from background values. Samples were digested in Teflon™ closed beakers, with a combination of HF, HCl and $HNO_3$, at 110 °C. Pb separation was performed using resin AG1X8™: matrix elution with HBr 0.8 N and Pb recovery with HCl 6 N, following the procedures described in Maccali et al. (2012) and Manhes et al. (1980)[67,68]. Nd was then separated following analytical procedure described in Maccali et al. (2013) and Song et al. (2022)[69,70]: digested samples were subjected to iron removal using AG1X8™ resin columns with 6 N HCl twice, replacing the resin each time. The samples were dried, treated with 14 N $HNO_3$, and dried again. Next, light rare earth elements (LREE) separation was performed using TRU SPEC™ resin. The matrix was eluted with 1 N $HNO_3$, and LREE were eluted and collected with 0.05 N $HNO_3$. The concentrated LREE samples underwent Nd separation using LN Spec™ resin. The first elution with 4 mL of 0.25 N HCl removes Ce, allowing Nd collection afterwards, with 2 mL of 0.25 N HCl. Nd and Pb isotope ratios ($^{143}Nd/^{144}Nd$, $^{207}Pb/^{206}Pb$) were analyzed on a Nu Plasma II™, a Multi-Collector Inductively Coupled Plasma Mass Spectrometer (MC-ICP-MS) with an Aridus™ II desolvating membrane. Replicate analyses of the standard JNdi-1 yielded mean values of $^{143}Nd/^{144}Nd = 0.512071 \pm 0.000029$ (2σ; $n = 15$), $0.512084 \pm 0.000051$ (2σ; $n = 11$) and $0.512079 \pm 0.000042$ (2σ; $n = 5$), in three batches of analyses. Results on JNdi-1 are slightly below the reference value ($0.512115 \pm 0.000007$)[71], as it is typical of Nd measurements on MC-ICP-MS due to mass bias after oxide formation in the plasma[72]. Therefore, results were normalised to 0.512115[71]. The external reproducibility (2σ) of $\varepsilon_{Nd(0)}$ values deduced from repeated measurements of the JNdi-1 standard was 0.8 $\varepsilon_{Nd(0)}$ units. Nd isotope ratios are expressed as $\varepsilon_{Nd(0)} = [(^{143}Nd/^{144}Nd)_{sample}/(^{143}Nd/^{144}Nd)_{CHUR}-1] \times 10000$, with the present-day $(^{143}Nd/^{144}Nd)_{CHUR}$ of 0.512638[73,74]. For Pb

measurements, instrumental mass bias was corrected using thallium (Tl) doping and internal normalisation to NBS 981 following Woodhead (2002)[75]. Reproducibility of Pb measurements was assessed using standard CGPB-1, which yielded $^{207}Pb/^{206}Pb$ of $0.844396 \pm 0.000039$ (2σ; $n = 6$). Procedural blanks were negligible compared with the sample size. Results are presented in Supplementary Dataset 4.

## Data availability

The radiogenic isotope data (neodymium and lead)[76], along with grain size measurements[77] (including sortable silt[78]) and quantitative X-ray diffraction data[79], have been archived in the Pangaea database under license CC-BY-4.0 and are also available in the supplementary datasets. The links to the data in Pangaea repository are: https://doi.pangaea.de/10.1594/PANGAEA.971292. https://doi.pangaea.de/10.1594/PANGAEA.971291. https://doi.pangaea.de/10.1594/PANGAEA.971295. https://doi.pangaea.de/10.1594/PANGAEA.971294.

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

## Acknowledgements

This work has been supported by the Natural Sciences and Engineering Research Council (NSERC) Canada Discovery Grant to CHM and JCMS. IMCS also acknowledges a post-doctoral fellowship from the Fonds de recherche du Québec – Nature et technologies (FRQ-NT). The authors thank A. Poirier and J. Gogot, both at Geotop, for the provided support in the laboratory.

## Author contributions

I.M.C.S. and C.H.M. conceived the study based on the palynology dataset provided by A.dV. and A.M.R.A. I.M.C.S. and J.-C.M.-S. generated the data, which were interpreted by I.M.C.S. and C.H.M. The paper was co-written by I.M.C.S., C.H.M., and A.dV.

## Competing interests

The authors declare no competing interests.
