## [Transparent Peer Review file · Nature Communications]

Cold spells over Greenland during the mid-Pliocene Warm Period

Corresponding Author: Dr Isabela Sousa

Version 0:

Reviewer comments:

Reviewer #1

(Remarks to the Author)

Dear Editor,

The manuscript by Sousa and colleagues investigates sediments collected from offshore southern Greenland to explore the glacial-interglacial characteristics of the ice sheet and regional oceanography across a time period in the middle Pliocene that is often thought to be an analogue for future climate change. In summary, using the geochemical composition of the sediments the authors use their results to question whether the use of this time period as an analogue is justified, and arrive at the conclusion that it is not. Based on their results, I agree with this. The work is of significance to the field of research, the methods are sound and well explained, and overall paper is packaged well. There are, of course, caveats to the timescales on this work and the ability to go between contemporary timescales and those of the middle Pliocene, but the authors have presented these issues clearly. I do not see any major issues in the text, it is largely well written and I believe the authors' interpretations and conclusions are fair and supported by the results. I have a few comments that are attached as tracked changes and some comments where I think there is room for some extra clarity and possibly some minor typographic errors - all of which are easily rectifiable. Overall, I do not see any major issues and I recommend this paper for publication.

If any of my comments are unclear, then the authors are welcome to contact me.

Best wishes,

Andrew Newton - Queen's University Belfast

Reviewer #2

(Remarks to the Author)

Manuscript summary

This paper presents high-resolution geochemical and grain size analysis of mid-Pliocene (3.2 – 3.0 Ma) sediments from IODP Site U1607 offshore southern Greenland. The site is located in Eirik Drift where sediments are delivered from East Greenland Current, influenced by the Western Boundary Undercurrent and presently receives IRD from eastern and southern Greenland iceberg-calving sources. Nd and Pb isotopes of the <63 µm (silt and clay) fraction aimed to distinguish between bedrock terranes from northern to southern Greenland, while grain size distributions were assessed for bottom current strength. Importantly, CaCO₃ was not removed for grain size analysis to include dolomite grains (deposited in hydrodynamic equilibrium) and biogenic CaCO₃, which were differentiated with quantitative XRD. Using a relative paleointensity-based magnetostratigraphic age model for U1307 tuned to U1308 by Blake-Mizen et al. (2019), the authors demonstrate clear ~41 kyr paced bi-modal sediment deposition for the mid-Pliocene (3.2 – 3.0 Ma).

While I consider the results of this study to be highly valuable, the manuscript is lacking in clarity and explanation, to the point that the interpretations are not directly supported by the results presented. In fact, I was required to read the Blake-Mizen et al. (2019) manuscript to gain enough understanding of the geographical and geological context of the site and region.

Discussion of concerns

- The depositional model is very confused and often feels contradicting. This is in part due to i) the paper structure (pertinent information to the deposition process and proxy response is scattered throughout the paper), ii) inconsistent use of terms

(e.g., glacials: used for glacial conditions and glacial cycles; grain size: used stand alone, or with sand, silt, sortable silt, SSmean, SS%), iii) references to conditions in Quaternary, Holocene, Miocene and Pliocene with no clear description of what these conditions are, and iv) a significant lack of explanation linking conditions to processes to proxy response. This could be easily remedied by a table or schematic diagram which should then be adequately described in-text.

- Throughout the manuscript, references seem to be used in place of explaining key background information that would strengthen the authors conclusions, here are three examples – but there are numerous instances where as a reader I am left unsure of the evidence or reasoning.

Line 88: "...possibly a stronger WBUC, as documented for post-Miocene interglacials 22." When is post-Miocene? How is this 'documented' ?

Line 89: "Increases in grain size, sand and SSmean are primarily linked to a stronger WBUC flow during interglacials24,33." By what means is a stronger WBUC flow linked to grain size?

Line 123: "During these warm intervals, surface conditions were variable38,39." How were they variable, what proxies, what conditions? And importantly, what is the significance of this statement on your results and interpretation?

- The assumption of CaCO₃ (differentiated from dolomite) as biogenic is concerning and reflects a broader lack of reference to or use of shipboard visual observations. Further, the use of continuous high-resolution measurements (e.g. color reflectance and physical properties) would greatly enhance this study for lithofacies and orbital assessment. However, if we take the authors assumption that CaCO₃ is biogenic, then Fig. S1, which shows a near to 1:1 (0.9x) relationship between CaCO₃ (%) and sortable silt mean size (µm), suggests the coarsening is not a response of increasing shear stress (bottom current strength) but purely the result of increased biogenic deposition. And to note the Fig. S1 formula for y is incorrect – it does not contain x.

- Regarding the grain size analysis, while the reader is buoyed by the consistency between the various measures (sand%, mean grain size, sortable silt and mean sortable silt), it i) appears that the use of different y-axis enables better comparable range (amplitude) of results are presented and ii) makes all four redundant – it would be better to have this in supplementary and one measure presented with (the background research) providing reasoning of why the specific proxy is used and how it is interpreted. Again, other available data e.g., IRD (Blake-Mizen et al., 2019) and/or physical property logs would provide more information.

- D18O data is poorly introduced, the mention of planktic is separated from specifying N. Atlantica which is solely used to refer to the data in the manuscript otherwise. This is all first presented in the discussion, when as previously published work, it should have been presented in the introduction.

- Of significant concern is the weak explanation for the Nd and Pb isotope. The results are not unique to the conclusions drawn – there is significant mixing of end members. The poor [science community] knowledge of Greenland geology is not adequately addressed. The modern values are the strongest line of evidence – to support the interpretation of interglacial conditions, but this is barely discussed (again taking by the place of a reference) and does not address the Iceland source which appears to be a significant influence of sedimentation today (Blake-Mizen et al., 2019). Fundamentally, explanations for why erosion and sediment delivery to the site during glacial conditions would be sourced by central Greenland zones is not apparent in the manuscript – which is central to the study. This is inhibited by a lack of introduction of the ice sheet models presented in Fig. 1. At no point are the climate conditions for the models presented – I would presume a glacial state and interglacial state to be represented but it is unclear what the differences between the models are. I cannot agree with the authors that the results exclude ice sheet scenarios.

- Figure captions need more detail.

The models in Fig 1b-c need a specific time or condition they are modelled for – are these peak mPWP interglacial conditions, glacial periods, early Pliocene, Late Pliocene? More geological/ geographical context should be added in the site map (e.g. other sites and currents).

Fig. 3 δ18O of N. atlantica needs to be more specific – Planktic (N. atlantica) δ18O stratigraphy from IODP Site U1307 27... again using references in place of explanation.

Fig 4 does not particularly add anything – there is no interpretation with this and is simply explained in text. Would prefer this was a sediment model.

Recommendations

All of the above concerns can be addresses with more detailed introduction including site location, modern deposition including all oceanography – only WBUC mentioned, published data, Greenland ice sheet history – particularly from ODP 646, ODP 907 and previous work from U1307. Simplification of grain size analysis for a very clear depositional model that explains the proxy, the process and environmental interpretation. The concerns about the CaCO₃ need to be addresses – but this should be as simple as using visual observation from shipboard descriptions. References should not be used in place of explanation, or require the reader to have read all of these references – the relevance to this study needs to be summarised.

However, the conclusions are not uniquely supported by the data presented. The results do not preclude ice sheet configurations as has been proposed here. The results have not provided adequate reasoning for changes in strength of the WBUC, with further interpretation of long-term trends for a 200 kyr record. Particularly, the focus on global orbital conditions during the mid-Pliocene as an inappropriate analogue is out of place - as stated in the abstract "... low overall summer

insolation conditions at high latitudes, a feature not anticipated in the near future... the mPWP interval cannot be seen as a genuine analogue specifically for predicting the fate of the GIS." A 1.2 Myr node in obliquity has been noted in numerous studies and is in fact a primary reason for studying the mPWP – to test the influence of precession – this is not new but is seemingly presented as such.

The weighting of various conclusions in the title, abstract and discussion are not consistent and leave the reader confused as to the point of the paper.

The very important result of obliquity pacing of sediments off Greenland is of significant value (can't overstate this enough!!!!) to address the question of climate pacing in Greenland when distal IRD records that have multiple sources are unable to distinguish this. Specifically, this highlights variability in polar ice sheet response and global records (EAIS – eccentricity; Patterson et al., 2014)(WAIS – obliquity; Naish et al., 2009)(Iceberg Alley – precession; Reilly et al., 2021)(sea-level – precession; Grant et al., 2019)(LR04 – eccentricity; Lisiecki and Raymo, 2005). While refining the deposition model is needed to convince the reader of the processes it reflects (bottom water currents or ice sheet position and erosion), either process paced by obliquity would be of interest and contribution to the community.

I hope the authors find this feedback useful. I am eager to see these results published. With some simplification and clarification a much better paper will emerge.

Kind regards,

Dr. Georgia Grant, GNS Science
New Zealand

Reviewer #3

(Remarks to the Author)

The authors have investigated geochemical and sedimentological changes in a core collected from the Eirik Drift near Greenland during the Pliocene, a time interval that is of interest to paleoclimate science both for its similarities and differences to the Earth's present climate state. New data include neodymium and lead isotope ratios of fine-fraction sediments (<63 μm) as well as % carbonate and grain size (including sortable silt) of bulk sediment. The observed excursions in Nd and Pb isotopes are interesting, as is the long-term drift observed in maximum Pb isotope values and both max and min of Nd isotope values.

However, the limitations of the age model and the uncertainty regarding whether specific isotope excursions occur during glacial or interglacial intervals limit the insight that can be gained from these data alone. Specifically, large gaps in the stable oxygen isotope record make it difficult to A) align with the global curve and/or B) assign glacial/interglacial intervals to the geochemical record. The authors do discuss these issues, but the addition of more $\text{d}18\text{O}$ observations to fill gaps in the original record from Sarnthein et al. (2009) could have placed the other geochemical records on a firmer foundation for discussion of glacial-interglacial climate variability and trends.

In addition, it is puzzling that in this study, the Nd and Pb isotope analyses were done on the entire <63 micron size fraction, when it has been previously documented that the clay fraction (<3 micron) and silt fraction (3 – 63 micron) from samples in this region are chemically distinct and reflect different processes and sources. It is not clear why the authors cited prior studies that used a similar isotope provenance approach (Colville et al., 2011; Reyes et al., 2014) as proof of concept, but then did not follow those methods. In the Colville et al. and Reyes et al. studies, workers physically separated the 3-63 micron fraction (silt) via Stokes-Law settling, and then performed geochemical analysis on the silt-only fraction. In contrast, the present study did not separate clay from silt prior to analysis, and the reason for this was not discussed in either the main text or the supplement. Colville et al. (2011) suggested that the coarser silt fraction is primarily derived from glacial erosion and supplied from Greenland, while the finer clay fraction can be supplied from much more distant sources. The reason for lumping/analyzing them together in the present study is not clear, and this difference in sample processing makes direct analogy to previously published results in this region problematic.

The authors make a good point regarding the unsuitability of the mPWP as an analogue for near-future Earth climate – the orbital oscillations for these two intervals are different, and their results suggest that obliquity in particular is important for the behavior of high-latitude ice. Overall, this study does present some very intriguing patterns and trends for an interesting time period, from a site proximal to a globally important ice sheet. The interpretations are not as strongly supported by the data as they might be given stronger age control, or at least enhanced ability to assign glacial vs. interglacial conditions to excursions in Nd and Pb isotopes and other sediment properties. In addition, these records are inherently indirect measures of Greenland Ice Sheet behavior, being located offshore and downstream of the ice itself, making their link to the object of study somewhat tenuous. There was not much discussion of how the marine records compared with terrestrial evidence, which might strengthen the discussion.

Version 1:

Reviewer comments:

Reviewer #1

(Remarks to the Author)

I am satisfied that the authors have addressed the concerns I raised during my review. I have also looked through the issues raised by the other reviewers and believe that the authors have addressed them as effectively as possible, within the limitations/caveats of the data that are available for this work. I do not see any major edits that the authors could implement to improve the paper. This is an interesting piece of work and I recommend publication.

Reviewer #2

(Remarks to the Author)

To the authors,

The results describing southwestern Greenland Ice Sheet dynamics conclude strong evidence for obliquity pacing of glacial-interglacial changes to the sediment delivery forming the Eirik Drift. Where periods of radiogenic Nd isotopes comparable to present day are interpreted as interglacials, while glacial periods are denoted by depleted Nd (in phase with $^{207}\text{Pb}/^{206}\text{Pb}$). These results are linked to erosion of Greenland bedrock where sediments derived from the central interior are consistent with glacial periods and interglacial sediments appear more mixed.

This study highlights an absence of Greenland Ice Sheet dynamics for the warmer past and will be a helpful addition to the community to further understanding of ice sheet configuration.

The previous remarks made have been adequately addressed by the authors.

Kind regards,

Dr. Georgia Grant

Point-by-point response to the reviewers' comments

Reviewer #1

Line	Comment/Suggestion (in red)	Action
27	Modelling studies indicated that during the mPWP, the Greenland Ice Sheet (GIS) extent may have been considerably	Corrected
34	mostly indirect as they are restricted to information from marine records off Greenland margin	Corrected
-	contourite drift molded by the Western Boundary Undercurrent (WBUC) and its formation can be tracked back to the Miocene. Radiogenic isotope composition of sediments from this drift has been proven as a suitable tracer for glacial landscape erosion during the Pleistocene and the Holocene. We thus use a similar approach here to document the occurrence of glacial activity in southern Greenland during the mPWP.	The paragraph was modified during the review and does not feature the passages that needed correction anymore.
-	As Eirik Ridge sediments are supplied by the WBUC (Colville et al. 2011; Innocent et al. 1997; McCartney 1992), their grain size	The sentence was modified during the review and does not feature the passages that needed correction anymore.
-	As illustrated on Fig. 1	-
68	performed on the <63 μm sediment fraction	Corrected
100	The quasi-absence of dolomite leads to the inference that	Corrected
-	reversals and $\delta^{18}\text{O}$ stratigraphy to set an age model due to highly variable	Corrected
-	As the duration of glacials remains unascertained, the corollary is that interglacials are also difficult to estimate.	The sentence was modified during the review and does not feature the passages that needed correction anymore.
137	Observing the long-term trends from ~ 3220 to ~ 2980 ka	The sentence was modified during the review and does not feature the passages that needed correction anymore.
-	fewer document ation of potential linkages with the obliquity cycle, a resolution not necessarily reached using current approaches such as the oxygen isotope stratigraphy and paleomagnetic reversal ages. These approaches were indeed used for the setting of variable age models for core U1307(Sarnthein et al. 2009; Channell, Hodell, and Curtis 2016), including the Blake-Mizen et al. age model(Blake-Mizen et al. 2019) referred to on Fig. 3. The oxygen isotope data of planktic foraminifera were initially reported by Sarnthein et al. (Sarnthein et al. 2009) and show $\delta^{18}\text{O}$	The paragraph was modified during the review and does not feature the passages that needed correction anymore.
-	age model resolution is insufficient for the higher-order timescale investigated here	The sentence was modified during the review and does not feature the passages that needed correction anymore.
185	at high latitudes (Lee and Poulsen 2008) that would allow	Corrected
199	the interval are currently based on hypothetical correlation using the available data.	Corrected

211	marked by a progressively	Corrected
215	Nd and Pb isotope	Corrected
231	Adding that the	Corrected

- 1) I think I understand what this means, that through time the dominant source of the sediments progressively shifts north? I think this more needs to be written more clearly exactly what it means as it is not clear - “erosion progressing” is somewhat ambiguous. Basically this needs more clearly explained for a multi-disciplinary journal like this.

You understood it correctly; we modified the sentence to make it clearer. The initial sentence “Thus, throughout the mPWP, the glacial advances resulted in erosion progressing from the Archean Block and Nagsuqtoqidian Mobile Belt to the Paleogene volcanic terranes in eastern Greenland (Fig. 1)” has been replaced by:

“Thus, throughout the mPWP, the southeastern Greenland ice mass occupying KMB/AB terranes expanded progressively northward from one glacial pulse to the next, possibly merging with ice from the northeastern mountainous PV/NMB area (Fig. 1B). Consequently, there is a clear trend of increasing overall contribution of the PV terrane towards the upper part of the record. During the cold spells, enhanced ice erosion mainly released detrital supplies from the Archean Block (AB) and Nagsuqtoqidian Mobile Belt (NMB) (Figs. 1B and 2). Conversely, during interglacials, ice erosion was likely restricted to the mountainous KMB-AB area of southeastern Greenland (Figs. 1B; 2). This would explain the narrow cluster of Pb and Nd isotope compositions of interglacial detrital supplies between three endmembers (KMB, AB/NMB and PV): the KMB and AB/NMB endmembers, linked to ice erosion, and the PV endmember linked to increased WBUC supplies from the Denmark Strait (Fig. 2).”

- 2) “Nevertheless, the recurrence of six, possibly seven (Fig. 3) climatic oscillations over the 2.80 to 3.22 Ma interval would be compatible with a mean duration of ~ 40 ka for each glacial-interglacial cycle, taking into account uncertainties about the age model (see below).” Room for clarity here as the timeline wording and younger date does not match the Fig. Are the climatic oscillations the pulses of sediment? Needs clearer wording.

We tried to clarify as follows:

“Nevertheless, the recurrence of six, possibly seven (Fig. 3) isotopic excursions over the 2280-3220 ka interval studied (grey stripes in Fig. 3) would be compatible with a mean duration of ~40 ka for each cold-warm cycle. However, the ~40 ka-periodicity is not clearly expressed when using the Blake-Mizen et al.²⁸ age model (supplementary material, Fig. S7). This can be due to both uncertainty of anchor ages, in particular those of paleomagnetic reversals, and biases linked to linear interpolation between these ages. Assigning the glacial pulses to specific glacial stages defined by the LR04 stack³⁷ throughout the interval constrained by the paleomagnetic anchor ages (supplementary material, Fig. S5) could be a way to define an age model, which would fit better with a ~40 ka cycle forcing (supplementary material, Fig. S5).”

- 3) Not sure on what the journal requirements are, but ka is usually seen as a date, not a duration (ka).
As the journal accepts both ka and kyr, we maintained ka to avoid changing some figures.
- 4) Myr? Depends on journal preferences for dates and durations.
Ibid. Ma is used.

Reviewer #2

- 1) The depositional model is very confused and often feels contradicting. This is in part due to **i)** the paper structure (pertinent information to the deposition process and proxy response is scattered throughout the paper), **ii)** inconsistent use of terms (e.g., glacials: used for glacial conditions and glacial cycles; grain size: used stand alone, or with sand, silt, sortable silt, SS_{mean}, SS%), **iii)** references to conditions in Quaternary, Holocene, Miocene and Pliocene with no clear description of what these conditions are, and **iv)** a significant lack of explanation linking conditions to processes to proxy response. This could be easily remedied by a table or schematic diagram which should then be adequately described in-text.

- i) We tried to improve the description of the sediment accumulation on Eirik drift, by adding to the corresponding paragraph in the introduction, which now reads as follows:

“Previous paleoceanographic studies suggest that the strength of the WBUC was strongly reduced during glacial intervals but exhibited high flow rates notably during Quaternary interglacials²⁰⁻²⁴. At this site and along the trajectory of the WBUC, grain-size data analyses of surface sediments show that silt outweighs the clay size fraction that may originate from remote sources, north of the Denmark Strait (Fig. 1). Notwithstanding the dominance of silt, the effective grain size of the sediment is sensitive to changes in the strength of the WBUC^{20,24,29}. Disregarding the IRD content²⁸, this behavior results in the deposition of coarser sediments during interglacials due to high shear stress and winnowing³⁰.”

- i) We added a new reference which further documents the relationship between the glacial/interglacial change in the WBUC strength and the sediment grain size on Eirik Drift (Davies Stow, and Nicholson 2021).
 - ii) With reference to grain size, the fractions of concern are more precisely mentioned in the revised version. We also made it clear that we mention glacial/interglacial intervals with reference to the corresponding sedimentary units, but to "cold spells" for climate conditions leading to ice expansion as documented in the present record, i.e., spanning the 2980-3220 ka time window.
 - iii) All references cited in the text now include explicit information about the paleoceanographic or paleoclimate conditions of concern.
 - iv) In Fig. 3, we added an arrow indicating “coarser sediments = stronger WBUC”. In the same figure, the grey bars indicate the cold spells.
- 2) Throughout the manuscript, references seem to be used in place of explaining key background information that would strengthen the authors conclusions, here are three examples – but there are numerous instances where as a reader I am left unsure of the evidence or reasoning. **i)** Line 88: “...possibly a stronger WBUC, as documented for post-Miocene interglacials 22.” When is post-Miocene? How is this ‘documented’? **ii)** Line 89: “Increases in grain size, sand and SSmean are primarily linked to a stronger WBUC flow during interglacials^{24,33}.” By what means is a stronger WBUC flow linked to grain size? **iii)** Line 123: “During these warm intervals, surface conditions were variable^{38,39}.” How were they variable, what proxies, what conditions? And importantly, what is the significance of this statement on your results and interpretation? **iv)** The assumption of CaCO₃ (differentiated from dolomite) as biogenic is concerning and reflects a broader lack of reference to or use of shipboard visual observations. **v)** Further, the use of continuous high-resolution measurements (e.g. color reflectance and physical properties) would greatly enhance this study for lithofacies and orbital assessment. However, if we take the authors assumption that CaCO₃ is biogenic, **vi)** then Fig. S1, which shows a near to 1:1 (0.9x) relationship between CaCO₃ (%) and sortable silt mean size (µm), suggests the coarsening is not a response of increasing shear stress (bottom current strength) but purely the result of increased biogenic deposition. **v)** And to note the Fig. S1 formula for y is incorrect – it does not contain x.

As mentioned in item iii) above, all references are now complemented with specific information about the features they address.

- i) Several studies document the intensification of the WBUC since the Miocene. We refer to one of these (Müller-Michaelis and Uenzelmann-Neben 2014) and have added a few words about their findings, from seismic profile studies. Post-Miocene refers to the Plio-Quaternary.
- ii) A stronger WBUC flow results in coarser grain deposition, which is seen in sortable silt percentage, sortable silt mean size and sand percentage. We added appropriate reference. The paragraph was restructured and now reads as follows:

The Nd-Pb isotopic excursions are preceded by peaking calcite abundance, of variable amplitude, as well as by high mean grain size, sortable-silt size (SSmean) and sand contents (Fig. 3). These features broadly highlight two

distinct sedimentary regimes linked to the strength of the WBUC. One is related to strong currents during warm intervals, in particular during their middle to late part²². The other is related to low flow rates during cold intervals. The interglacial layers illustrated in Fig. 3 record an early stage with relatively fine material, followed by a late stage marked by sediment coarsening, which is a feature observed during the present interglacial in the core site vicinity²². The quasi-absence of dolomite leads to the inference that calcite peaking values record enhanced biogenic carbonate fluxes during the late part of the interglacial intervals. A similar feature characterizes the present interglacial²².

- iii) More information is now provided about the proxies used for reconstructing surface conditions during the warm intervals (planktic foraminifera assemblages and abundances).
- iv) Hillaire-Marcel et al. (1994) as well as Dmitrenko, Sivkov, and Rusakov (2009) have shown that biogenic calcite fluxes in the region are linked to an enhanced coccolithophore production and occur in the <10 μm fraction. Additionally, the sand-sized fraction measurements of Blake-Mizen et al. (2019) were performed after removal of carbonates. Their peaking percentages of the ≥ 63 μm sediment fraction are coeval with our peaks of mean grain size (μm), sand (%), SS_{mean} (μm) and SS (%) (see Supplementary Material, Fig. S3).
- v) We added color reflectance, magnetic susceptibility and NGR data to the Fig. S3 in the supplementary material.
- vi) Fig. S1 illustrates the fact that biogenic micrite calcite production increases during the late warm intervals marked by an enhanced WBUC.
- vii) The peaking calcite concentrations (red dots in Fig. S1) plot outside the 1:1 trend defined by calcite vs. SS_{mean}, meaning that the anomalous calcite contents are not accompanied by coarsening.

Figure S1: This figure was updated in the supplementary material.

- 3) Regarding the grain size analysis, while the reader is buoyed by the consistency between the various measures (sand%, mean grain size, sortable silt and mean sortable silt), **i)** it appears that the use of different y-axis enables better comparable range (amplitude) of results are presented and makes all four redundant – it would be better to have this in supplementary and one measure presented with (the background research) providing reasoning of why the specific proxy is used and how it is interpreted. **ii)** Again, other available data e.g., IRD (Blake-Mizen et al., 2019) and/or physical property logs would provide more information.

- i) We agree that all grain size parameters illustrated in Fig. 3 follow nearly similar patterns and may be considered redundant. We thus only plot the SS_{mean} and the mean grain size in the main figure, whereas their correlation with SS (%) and sand (%) remains documented in the supplementary material (Fig. S2).

Figure S2. Correlation between SS_{mean} (μm) and SS(%), and mean grain size (μm) and sand (%).

- ii) We added IRD from (Blake-Mizen et al. 2019) and shipboard measurements (natural gamma radiation, magnetic susceptibility and color reflectance) to the supplementary material. Blake-Mizen et al.'s IRD data are split in two fractions: $\geq 63 \mu\text{m}$ and $\geq 212 \mu\text{m}$. The fraction $\geq 63 \mu\text{m}$ mostly includes the sand-size fraction; its relative abundance does not necessarily represent the IRD flux vs other sedimentary supplies, but mostly the winnowing impact of an intensified WBUC concentrating coarse fractions and removing the fines. This may be illustrated by the present day WBUC velocities of up to $\sim 30 \text{ cm}\cdot\text{s}^{-1}$ (Holliday et al. 2007).
- 4) $\delta^{18}\text{O}$ data is poorly introduced, the mention of planktic is separated from specifying *N. Atlantica* which is solely used to refer to the data in the manuscript otherwise. This is all first presented in the discussion, when as previously published work, it should have been presented in the introduction.

Unfortunately, there is no continuous $\delta^{18}\text{O}$ record available for Site U1307. The existing *N. atlantica* data from Sarnthein et al. (2009) exhibit significant gaps. We mention this problem in the introduction and further elaborate on the importance of these data despite the impossibility to use them for the setting of an age model. The $\delta^{18}\text{O}$ data may still help developing correlations with the LR04 stack (Lisiecki and Raymo 2005). However, as now documented in the supplementary material of the revised version (Fig. 3 and Fig. S4), we tentatively set a closer correlation of the sequence analyzed here with the LR04 stack, based on the glacial/interglacial succession within the paleomagnetic intervals investigated.

- 5) **i)** Of significant concern is the weak explanation for the Nd and Pb isotope. **ii)** The results are not unique to the conclusions drawn – there is significant mixing of endmembers. The poor [science community] knowledge of Greenland geology is not adequately addressed. **iii)** The modern values are the strongest line of evidence to support the interpretation of interglacial conditions, but this is barely discussed (again taking by the place of a reference) and **iv)** does not address the Iceland source which appears to be a significant influence of sedimentation today (Blake-Mizen et al., 2019). **v)** Fundamentally, explanations for why erosion and sediment delivery to the site during glacial conditions would be sourced by central Greenland zones is not apparent in the manuscript – which is central to the study. **vi)** This is inhibited by a lack of introduction of the ice sheet models presented in Fig. 1. At no point are the climate conditions for the models presented – I would presume a glacial state and interglacial state to be represented but it is unclear what the differences between the models are. I cannot agree with the authors that the results exclude ice sheet scenarios.

- i) We added a new section in the supplementary material addressing the geology of Greenland terranes and discussing the meaning of the Nd and Pb signatures.
 - ii) As illustrated in Fig. 2, three/four endmembers are variably mixed. Previous works have modelled the more recent contribution of different terranes to a nearby site (Reyes et al. 2014; Fagel and Hillaire-Marcel 2006) and showed that unradiogenic Nd primarily originates from Archean sources in southern Greenland and can thus be used as a tracer of related inland ice erosion in this area.
 - iii) We added a couple of sentences in the revised to clarify this aspect. The modern value provide evidence for some PV (Paleogene volcanism) source material supplies with the modern strong WBUC, either from direct erosion through the Greenland Strait, or from ice calving in ice terminals of NE Greenland, as Greenland is presently fully glaciated. This was not the case during the Pliocene, prior to the “intensification of the Northern Hemisphere Glaciation” at ~2.7 Ma (Abell et al. 2021; Naafs et al. 2010; Haug et al. 1999)
 - iv) The Iceland source is within the terrane called PV (Paleogene volcanism) as well as the northeastern Greenland source, as documented in the Blake-Mizen et al. (2019) paper. PV is composed of basalts ($\epsilon_{Nd(0)}$ up to +4). Sediments from this source thus drive $\epsilon_{Nd(0)}$ towards higher values, but do not provide a means to distinguish Greenland vs. Iceland supplies.
 - v) The current understanding of Greenland terranes (e.g., MacGregor et al., 2023) enabled us to identify, south to north, the Ketilidian Mobile Belt (KMB), the Archean Block (AB), Nagssugtoqidian Mobile Belt (NMB), and the Paleogene volcanism (PV). During the mPWP glacials we see enhanced source supplies not from "central" Greenland (NMB and PV), but rather from southeastern Greenland (KMB/AB). Worth to mention is the fact that ice streams transport material to the sea mostly only over a few km (see Paulen et al., R.C. and McMartin, I.E. 2009)). Assuming a weak WBUC during glacials, one would not expect significant long-distance transport of glacially eroded material, at least in coarse fractions.
 - vi) We tried to clarify the model-based possible distributions of ice during the mPWP in the introduction. Figs. 1B,C show two models of ice masses during the Pliocene. In one scenario, ice is restricted to the mountains in south and east Greenland (Fig. 1B) and in the other, ice is restricted to northern Greenland (Fig. 1C). Our data ($\epsilon_{Nd(0)}$ and $^{207}Pb/^{206}Pb$) are compatible with scenario 1B, i.e., as the sediment composition during warm intervals is mostly controlled by two endmembers: Paleogene volcanics and the KMB terrane. These intervals were the intervals with minimal ice cover.
- 6) **i)** The models in Fig 1b-c need a specific time or condition they are modelled for – are these peak mPWP interglacial conditions, glacial periods, early Pliocene, Late Pliocene? **ii)** More geological/ geographical context should be added in the site map (e.g. other sites and currents). **iii)** Fig. 3 $\delta^{18}O$ of *N. atlantica* needs to be more specific – Planktic (*N. atlantica*) $\delta^{18}O$ stratigraphy from IODP Site U1307 27... again using references in place of explanation. **iv)** Fig 4 does not particularly add anything – there is no interpretation with this and is simply explained in text. **v)** Would prefer this was a sediment model.
- i) Figs. 1B,C are ice sheet surface elevation model outputs with minimal ice cover during the warmer part of the Pliocene (~ 3.26-3.02 Ma; Koenig et al. 2015). We added more information about each model in the revised version.
 - ii) The most important currents (in the context of our research), likely active during the mPWP, are indicated in the schematic map of Fig. 1. More detailed information about currents during this interval would be largely speculative.
 - iii) We improved the information about *N. atlantica*, with relevant data from U1307.
 - iv) Although the relationship between obliquity nodes and cooling trends has been mentioned in general terms (e.g., O’Neill and Broccoli 2021), we discuss it here with specific reference to the mPWP interval.

It constitutes an important element that may help understanding why ice could grow under the high $p\text{CO}_2$ of the interval. As such an orbital situation cannot be expected in the near future, we think that the mPWP interval cannot be considered as a real analogue of the near future of the ice cover of Greenland. Thus, we would like to keep the Fig. 4, because this element is discussed in the text. It seems uneasy to infer a sediment model for Eirik Drift during the mPWP based on the scarce information existing.

- v) Unfortunately, we cannot add to the sedimentological studies of site U1307 cores (Sarnthein et al. 2009; Blake-Mizen et al. 2019) and present a comprehensive sediment model, because a figure combining the Ice-Sheet distribution over Greenland and the WBUC strength would be very speculative given the data we have.
- 7) **i)** All of the above concerns can be addressed with more detailed introduction including site location, modern deposition including all oceanography – only WBUC mentioned, published data, Greenland ice sheet history – particularly from ODP 646, ODP 907 and previous work from U1307. **ii)** Simplification of grain size analysis for a very clear depositional model that explains the proxy, the process and environmental interpretation. **iii)** The concerns about the CaCO_3 need to be addressed – but this should be as simple as using visual observation from shipboard descriptions. **iv)** References should not be used in place of explanation, or require the reader to have read all of these references – the relevance to this study needs to be summarised.
- i) The introduction has been updated to include more details, as suggested, and several passages lacking explanation, are now better developed. About Greenland Ice Sheet history, most studies focus on the onset of the Northern Hemisphere Glaciation (long-term cooling of Greenland) or have no comparable temporal resolution. This is a gap in knowledge that our work is filling. Apart from that, the models for the GIS in the Pliocene are the ones cited on Figs. 1B,C, the captions of which were improved.
- ii) Fig. 3 was improved to include information about current strength and clarify the link between the results and the oceanographic interpretation.
- iii) We further discussed the CaCO_3 content and added shipboard measurements to the supplementary material. There are no smear slides from the targeted interval, but $\text{IDR} > 63 \mu\text{m}$ form (Blake-Mizen et al. 2019) were measured after carbonate removal and depict peaks coeval with our coarse fraction contents. These data were added to the supplementary material.
- iv) We reviewed all references to make sure they are not used without explanation about their scientific pertinence.
- 8) However, **i)** the conclusions are not uniquely supported by the data presented. **ii)** The results do not preclude ice sheet configurations as has been proposed here. The results have not provided adequate reasoning for changes in strength of the WBUC, with further interpretation of long-term trends for a 200 ka record. **iii)** Particularly, the focus on global orbital conditions during the mid-Pliocene as an inappropriate analogue is out of place - as stated in the abstract "... low overall summer insolation conditions at high latitudes, a feature not anticipated in the near future... the mPWP interval cannot be seen as a genuine analogue specifically for predicting the fate of the GIS." A 1.2 Myr node in obliquity has been noted in numerous studies and is in fact a primary reason for studying the mPWP – to test the influence of precession – this is not new but is seemingly presented as such.
- i) In the revised version, we think we highlight better how our conclusions are supported by the data (periodic erosion of Precambrian terranes = ice advances, coarser sediments = stronger current). We also improved the discussion about why our data are more compatible with one of the ice-sheet configurations. The strength of the WBUC is inferred by the SS_{mean} and $\text{SS}(\%)$, while erosion of Precambrian terranes is inferred from radiogenic isotope signatures. Our data do not allow us to clearly infer what could have been the drivers of WBUC changes, but we added a reference that discusses a southward shift of the deep-water formation regions during "glacials" (implying that the main North Atlantic Deep Water route affected Eirik Drift only during warm intervals) (Müller-Michaelis and Uenzelmann-Neben 2014).
- ii) Indeed, the potential role of low obliquity conditions for ice growth has been evoked in particular by O'Neill and Broccoli (2021) and we do not claim any originality in mentioning it. However, as

mentioned above, the very low obliquity values of the mPWP interval may account for the ice growth pulses over Greenland, despite the high pCO₂ value of the time. We think important to mention this aspect in the abstract, especially with respect to the absence of such orbital conditions in the near future of the Earth, whereas several papers refer to the mPWP as a potential analogue of the future of the planet. We added a reference to the O'Neill and Broccoli (2021) paper to clarify this aspect.

- 9) **i)** The weighting of various conclusions in the title, abstract and discussion are not consistent and leave the reader confused as to the point of the paper. **ii)** The very important result of obliquity pacing of sediments off Greenland is of significant value (can't overstate this enough!!!) to address the question of climate pacing in Greenland when distal IRD records that have multiple sources are unable to distinguish this. Specifically, this highlights variability in polar ice sheet response and global records (EAIS – eccentricity; Patterson et al., 2014)(WAIS – obliquity; Naish et al., 2009)(Iceberg Alley – precession; Reilly et al., 2021)(sea-level – precession; Grant et al., 2019)(LR04 – eccentricity; Lisiecki and Raymo, 2005). **iii)** While refining the deposition model is needed to convince the reader of the processes it reflects (bottom water currents or ice sheet position and erosion), either process paced by obliquity would be of interest and contribution to the community.
- i) We restructured several paragraphs to emphasize the periodic enhanced glacial erosion of Archean terranes under high pCO₂ conditions of the mPWP, which is the central finding of our work. Nonetheless, our data also allow us to discuss the strength of WBUC based on the grain size measurements.
 - ii) Indeed, the obliquity pacing is of significant value, but due to uncertainties on the age model, we tried to discuss it with some caution and even added a possible way to revise the chronostratigraphy through a direct correlation of our glacial/interglacial sequence with the LR04 stack spanning the same paleomagnetic boundaries (see supplementary material, Fig. SX)
 - iii) We did not include a diagram with the depositional model because the distribution of the ice mass outlets, as well as the morphology of Eirik Drift are not well documented for the mPWP interval.

Reviewer #3

- 1) Specifically, large gaps in the stable oxygen isotope record make it difficult to A) align with the global curve and/or B) assign glacial/interglacial intervals to the geochemical record. The authors do discuss these issues, but the addition of more d18O observations to fill gaps in the original record from Sarnthein et al. (2009) could have placed the other geochemical records on a firmer foundation for discussion of glacial-interglacial climate variability and trends.

The gaps in the δ¹⁸O records are due to the lack of foraminifera in some parts of the cored sequences. Unfortunately, to our best knowledge, there are no other available records with the temporal resolution of our own study. Nonetheless, we added shipboard measurements to the supplementary material and tried at a direct correlation of our glacial/interglacial sequence with the LR04 stack spanning the same paleomagnetic boundaries (see supplementary material, Fig. S5).

- 2) In addition, it is puzzling that in this study, the Nd and Pb isotope analyses were done on the entire <63 micron size fraction, when it has been previously documented that the clay fraction (<3 micron) and silt fraction (3 – 63 micron) from samples in this region are chemically distinct and reflect different processes and sources. It is not clear why the authors cited prior studies that used a similar isotope provenance approach (Colville et al., 2011; Reyes et al., 2014) as proof of concept, but then did not follow those methods. In the Colville et al. and Reyes et al. studies, workers physically separated the 3-63 micron fraction (silt) via Stokes-Law settling, and then performed geochemical analysis on the silt-only fraction. In contrast, the present study did not separate clay from silt prior to analysis, and the reason for this was not discussed in either the main text or the supplement. Colville et al. (2011) suggested that the coarser silt fraction is primarily derived from glacial erosion and supplied from Greenland, while the finer clay fraction can be supplied from much more distant sources. The reason for lumping/analyzing them together in the present study is not clear, and this difference in sample processing makes direct analogy to previously published results in this region problematic.

It is unfortunate that there is no consensus in the community about the proper size to use for the documenting of sediment sources of hemipelagic sediments from radiogenic isotope measurements. Here, the silt fraction was not separated from the clay as grain size analyses showed that clay constitutes only 10% of the overall sediment. In a very conservative estimate, in which clays from the cold spells would have been entirely derived from Precambrian sources ($^{143}\text{Nd}/^{144}\text{Nd}=0.510978$; Reyes et al. 2014), their presence would account for a maximum increment of $\sim 1 \epsilon_{\text{Nd}(0)}$ units, meaning that excursions would still be distinguishable from background values. We added this information and discussion to the supplementary material. Thus, the Nd and Pb isotopic signatures from the $< 63 \mu\text{m}$ sediment fraction are thought representative of the mixing of sediments from proximal glacial erosion and long-distance transport of fine particles, this, in particular during interglacials, when the WBUC was stronger.

- 3) There was not much discussion of how the marine records compared with terrestrial evidence, which might strengthen the discussion.

Very few terrestrial data are available for the targeted timeframe in Greenland, especially with high temporal resolution. We mentioned Csank et al. 2011), who estimated land surface temperatures and suggested temperatures ~ 11 to 16°C warmer than present. We added a new reference (Bennike et al. 2002), who show that seawater temperatures were higher than today during the Pliocene (although the work has not the same temporal resolution as the one presented in this study). A similar conclusion can be made with respect to the very recent publication of Pang et al. (2024) who tries to document paleosea-surface conditions between 3.65–3.37 Ma, i.e., prior to the mPWP interval.

References

- Abell, Jordan T., Gisela Winckler, Robert F. Anderson, and Timothy D. Herbert. 2021. "Poleward and Weakened Westerlies during Pliocene Warmth." *Nature* 589 (7840): 70–75. <https://doi.org/10.1038/s41586-020-03062-1>.
- Bennike, Ole, Niels Abrahamsen, Małgorzata Bak, Carsten Israelson, Peter Konradi, Jens Matthiessen, and Andrzej Witkowski. 2002. "A Multi-Proxy Study of Pliocene Sediments from Île de France, North-East Greenland." *Palaeogeography, Palaeoclimatology, Palaeoecology* 186 (1–2): 1–23. [https://doi.org/10.1016/S0031-0182\(02\)00439-X](https://doi.org/10.1016/S0031-0182(02)00439-X).
- Blake-Mizen, Keziah, Robert G. Hatfield, Joseph S. Stoner, Anders E. Carlson, Chuang Xuan, Maureen Walczak, Kira T. Lawrence, James E.T. Channell, and Ian Bailey. 2019. "Southern Greenland Glaciation and Western Boundary Undercurrent Evolution Recorded on Eirik Drift during the Late Pliocene Intensification of Northern Hemisphere Glaciation." *Quaternary Science Reviews* 209 (April):40–51. <https://doi.org/10.1016/j.quascirev.2019.01.015>.
- Channell, J.E.T., D.A. Hodell, and J.H. Curtis. 2016. "Relative Paleointensity (RPI) and Oxygen Isotope Stratigraphy at IODP Site U1308: North Atlantic RPI Stack for 1.2–2.2 Ma (NARPI-2200) and Age of the Olduvai Subchron." *Quaternary Science Reviews* 131 (January):1–19. <https://doi.org/10.1016/j.quascirev.2015.10.011>.
- Colville, Elizabeth J., Anders E. Carlson, Brian L. Beard, Robert G. Hatfield, Joseph S. Stoner, Alberto V. Reyes, and David J. Ullman. 2011. "Sr-Nd-Pb Isotope Evidence for Ice-Sheet Presence on Southern Greenland During the Last Interglacial." *Science* 333 (6042): 620–23. <https://doi.org/10.1126/science.1204673>.
- Csank, Adam Z., Aradhna K. Tripathi, William P. Patterson, Robert A. Eagle, Natalia Rybczynski, Ashley P. Ballantyne, and John M. Eiler. 2011. "Estimates of Arctic Land Surface Temperatures during the Early Pliocene from Two Novel Proxies." *Earth and Planetary Science Letters* 304 (3–4): 291–99. <https://doi.org/10.1016/j.epsl.2011.02.030>.
- Davies, Sally, Dorrik Stow, and Uisdean Nicholson. 2021. "Late Glacial to Holocene Sedimentary Facies of the Eirik Drift, Southern Greenland Margin: Spatial and Temporal Variability and Paleoceanographic Implications." *Marine Geology* 440 (October):106568. <https://doi.org/10.1016/j.margeo.2021.106568>.
- Dmitrenko, O. B., V. V. Sivkov, and V. Yu. Rusakov. 2009. "Late Quaternary Migrations of the Subarctic Front in the North Atlantic (Based on Lithology and Nannofossils)." *Oceanology* 49 (2): 242–56. <https://doi.org/10.1134/S000143700902009X>.
- Fagel, Nathalie, and Claude Hillaire-Marcel. 2006. "Glacial/Interglacial Instabilities of the Western Boundary Under Current during the Last 365 Ka from Sm/Nd Ratios of the Sedimentary Clay-Size Fractions at ODP Site 646 (Labrador Sea)." *Marine Geology* 232 (1–2): 87–99. <https://doi.org/10.1016/j.margeo.2006.08.006>.

- Fagel, Nathalie, Claude Hillaire-Marcel, and Christian Robert. 1997. "Changes in the Western Boundary Undercurrent Outflow since the Last Glacial Maximum, from Smectite/Illite Ratios in Deep Labrador Sea Sediments." *Paleoceanography* 12 (1): 79–96. <https://doi.org/10.1029/96PA02877>.
- Haug, Gerald H., Daniel M. Sigman, Ralf Tiedemann, Thomas F. Pedersen, and Michael Sarnthein. 1999. "Onset of Permanent Stratification in the Subarctic Pacific Ocean." *Nature* 401 (6755): 779–82. <https://doi.org/10.1038/44550>.
- Hillaire-Marcel, C., A. De Vernal, G. Bilodeau, and G. Wu. 1994a. "Isotope Stratigraphy, Sedimentation Rates, Deep Circulation, and Carbonate Events in the Labrador Sea during the Last ~ 200 Ka." *Canadian Journal of Earth Sciences* 31 (1): 63–89. <https://doi.org/10.1139/e94-007>.
- Holliday, N. Penny, Amélie Meyer, Sheldon Bacon, Steven G. Alderson, and Beverly De Cuevas. 2007. "Retroflexion of Part of the East Greenland Current at Cape Farewell." *Geophysical Research Letters* 34 (7): 2006GL029085. <https://doi.org/10.1029/2006GL029085>.
- Innocent, Christophe, Nathalie Fagel, Ross K. Stevenson, and Claude Hillaire-Marcel. 1997. "Sm–Nd Signature of Modern and Late Quaternary Sediments from the Northwest North Atlantic: Implications for Deep Current Changes since the Last Glacial Maximum." *Earth and Planetary Science Letters* 146 (3–4): 607–25. [https://doi.org/10.1016/S0012-821X\(96\)00251-8](https://doi.org/10.1016/S0012-821X(96)00251-8).
- Koenig, S. J., A. M. Dolan, B. De Boer, E. J. Stone, D. J. Hill, R. M. DeConto, A. Abe-Ouchi, et al. 2015. "Ice Sheet Model Dependency of the Simulated Greenland Ice Sheet in the Mid-Pliocene." *Climate of the Past* 11 (3): 369–81. <https://doi.org/10.5194/cp-11-369-2015>.
- Lee, S.-Y., and C. J. Poulsen. 2008. "Amplification of Obliquity Forcing through Mean Annual and Seasonal Atmospheric Feedbacks." *Climate of the Past* 4 (4): 205–13. <https://doi.org/10.5194/cp-4-205-2008>.
- Lisiecki, Lorraine E., and Maureen E. Raymo. 2005. "A Pliocene-Pleistocene Stack of 57 Globally Distributed Benthic $\delta^{18}\text{O}$ Records: PLIOCENE-PLEISTOCENE BENTHIC STACK." *Paleoceanography* 20 (1): n/a-n/a. <https://doi.org/10.1029/2004PA001071>.
- McCartney, M.S. 1992. "Recirculating Components to the Deep Boundary Current of the Northern North Atlantic." *Progress in Oceanography* 29 (4): 283–383. [https://doi.org/10.1016/0079-6611\(92\)90006-L](https://doi.org/10.1016/0079-6611(92)90006-L).
- Müller-Michaelis, Antje, and Gabriele Uenzelmann-Neben. 2014. "Development of the Western Boundary Undercurrent at Eirik Drift Related to Changing Climate since the Early Miocene." *Deep Sea Research Part I: Oceanographic Research Papers* 93 (November):21–34. <https://doi.org/10.1016/j.dsr.2014.07.010>.
- Naafs, B. David A., Ruediger Stein, Jens Heffer, Nabil Khélifi, Stijn De Schepper, and Gerald H. Haug. 2010. "Late Pliocene Changes in the North Atlantic Current." *Earth and Planetary Science Letters* 298 (3–4): 434–42. <https://doi.org/10.1016/j.epsl.2010.08.023>.
- O'Neill, Grainne R., and Anthony J. Broccoli. 2021. "Orbital Influences on Conditions Favorable for Glacial Inception." *Geophysical Research Letters* 48 (21): e2021GL094290. <https://doi.org/10.1029/2021GL094290>.
- Pang, Xiaolei, Antje Helga Luise Voelker, Sihua Lu, and Xuan Ding. 2024. "Distinct Seasonal Changes and Precession Forcing of Surface and Subsurface Temperatures in the Mid-Latitudinal North Atlantic during the Onset of the Late Pliocene." <https://doi.org/10.5194/egusphere-2024-603>.
- Paulen, R.C., and McMartin, I.E. 2009. "Ice-Flow Indicators and the Importance of Ice-Flow Mapping for Drift Prospecting." In *Application of Till and Stream Sediment Heavy Mineral and Geochemical Methods to Mineral Exploration in Western and Northern Canada*. Geological Association of Canada.
- Reyes, Alberto V., Anders E. Carlson, Brian L. Beard, Robert G. Hatfield, Joseph S. Stoner, Kelsey Winsor, Bethany Welke, and David J. Ullman. 2014. "South Greenland Ice-Sheet Collapse during Marine Isotope Stage 11." *Nature* 510 (7506): 525–28. <https://doi.org/10.1038/nature13456>.
- Sarnthein, M., G. Bartoli, M. Prange, A. Schmittner, B. Schneider, M. Weinelt, N. Andersen, and D. Garbe-Schönberg. 2009. "Mid-Pliocene Shifts in Ocean Overturning Circulation and the Onset of Quaternary-Style Climates." *Climate of the Past* 5 (2): 269–83. <https://doi.org/10.5194/cp-5-269-2009>.

Isabela Moreno Cordeiro de Sousa
Geotop, UQAM
201, Président-Kennedy Ave. 7th floor, room PK-7150
Montréal, QC, H2X 3Y7
moreno_cordeiro_de_sousa.isabela@uqam.ca

18 January 2025

Dear Reviewers,

On behalf of my co-authors, I would like to thank you for your thorough review on our manuscript Cold spells over Greenland during the mid-Pliocene Warm Period. Your comments and suggestions have greatly improved the quality and clarity of our work.

We truly appreciate the time and effort you dedicated to this review process.

Sincerely,

Isabela Moreno Cordeiro de Sousa